# A Pan-Global Study of Bacterial Leaf Spot of Chilli Caused by *Xanthomonas* spp.

**DOI:** 10.3390/plants11172291

**Published:** 2022-09-01

**Authors:** Desi Utami, Sarah Jade Meale, Anthony Joseph Young

**Affiliations:** 1School of Agriculture and Food Sciences, Faculty of Science, The University of Queensland, Brisbane 4343, Australia; 2Department of Agricultural Microbiology, Faculty of Agriculture, Universitas Gadjah Mada, Yogyakarta 55281, Indonesia

**Keywords:** bacterial leaf spot, chilli, *Xanthomonas* spp., disease detection

## Abstract

Bacterial Leaf Spot (BLS) is a serious bacterial disease of chilli (*Capsicum* spp.) caused by at least four different *Xanthomonas* biotypes: *X. euvesicatoria* pv. *euvesicatoria*, *X. euvesicatoria* pv. *perforans*, *X. hortorum* pv. *gardneri*, and *X. vesicatoria*. Symptoms include black lesions and yellow halos on the leaves and fruits, resulting in reports of up to 66% losses due to unsalable and damaged fruits. BLS pathogens are widely distributed in tropical and subtropical regions. *Xanthomonas* is able to survive in seeds and crop residues for short periods, leading to the infections in subsequent crops. The pathogen can be detected using several techniques, but largely via a combination of traditional and molecular approaches. Conventional detection is based on microscopic and culture observations, while a suite of Polymerase Chain Reaction (PCR) and Loop-Mediated Isothermal Amplification (LAMP) assays are available. Management of BLS is challenging due to the broad genetic diversity of the pathogens, a lack of resilient host resistance, and poor efficacy of chemical control. Some biological control agents have been reported, including bacteriophage deployment. Incorporating stable host resistance is a critical component in ongoing integrated management for BLS. This paper reviews the current status of BLS of chilli, including its distribution, pathogen profiles, diagnostic options, disease management, and the pursuit of plant resistance.

## 1. Introduction

Chilli (*Capsicum* spp.) is an important spice or condiment which is grown in most countries in the world [1]. It is also iconic and has made significant impacts in popular and traditional culture. Also called capsicum, bell pepper and chilli pepper, it originated in tropical regions of Latin America [2]. Chilli is believed to be the first domesticated spice crop [3]. Although it is a quintessential element of the cuisine in many European and Asian nations, chilli is only a relatively recent acquisition.

The year 1492 saw the voyage of Christopher Columbus which opened an exchange of population, concepts, and food crops between what was termed the New World and the Old World. A significant part of the Columbian Exchange, chilli was one of the first crops introduced into Europe from the Americas [4]. At the time there were significant economic interests in black pepper, which saw its price exceed that of gold [5]. Thus, with the poor supply of black pepper, the flavor impact of chilli saw it become one of the world’s most popular condiments [6].

The most recognizable attribute of chilli is its spicy sensation. This sensation is considered one of six main tastes, along with sweet, bitter, sour, salty, and umami [4]. The spiciness of chilli is conferred by its natural alkaloids, known as capsaicinoids, which give the sensation of heat. There are two major capsaicinoids: capsaicin and dihydrocapsaicin, representing approximately 80–90% of the total capsaicinoids in most chilli [7]. It is thought that the capsaicinoids evolved as a way to prevent mammals from eating the fruit, while allowing birds, with their greater potential for seed transport, to be the primary agents for dispersal [8]. It is ironic that this chemical means to dissuade mammals has delivered a sought-after relish which has seen chillis transported and cultivated worldwide by humans.

While chilli fruits are primarily considered a spice, they have many other uses. They can be used for fresh and processed fruits, medicinal ingredients [9], food dyes [10], and even as insecticides [11]. Apart from its flavor, chilli has rich nutritional values, specifically high levels of vitamins A, B, C, and E [12,13]. The popularity of chilli has seen global production steadily increase. The Food and Agriculture Organization [14] reported that chilli production has increased from 29.6 million tons/year in 2010 to 40.3 million tons/year in 2020. Chilli was estimated to have approximately US $30 billion and US $3.8 billion in global value for fresh and dried chilli crops, respectively [15].

## 2. Bacterial Leaf Spot

A wide range of biotic and abiotic factors affect chilli production. Abiotic factors, such as temperature, drought, flooding, and salinity [16], largely define where chillis are optimally propagated. The presence of biotic factors, such as diseases and pests, can have major effects on chilli yields across cultivars. One of the most economically threatening diseases of chilli is Bacterial Leaf Spot (BLS). BLS is caused by at least four different *Xanthomonas* biotypes: *X. euvesicatoria* pv. *euvesicatoria*, *X. euvesicatoria* pv. *perforans*, *X. hortorum* pv. *gardneri* and *X. vesicatoria*. However, it is possible that undiscovered species may also be present. These *Xanthomonas* biotypes infect plants from the Solanaceae family, including capsicum (*Capsicum annuum*) and tomato (*Lycopersicon esculentum*) [17]. The Gram-negative, motile, aerobic, short rod-shaped bacteria can infect leaves, fruits, and stems, causing necrotic lesions and defoliation [18], as shown in Figure 1.

BLS-associated leaf lesions are often surrounded by a yellow halo of variable size. Approximately 24 h post-inoculation, hypersensitive cultivars exhibit a tissue collapse reaction from dark green to grey, while susceptible reactions lead to the appearance of water-soaked lesions, 3 days after inoculation [17]. On some cultivars, leaves infected with *Xanthomonas* may display several small lesions (1 mm) which can cover >80% of the leaf area, whereas in other cultivars, fewer large lesions (>5 mm) may be found. BLS symptoms on chilli leaves and fruits are presented in Figure 2. BLS causes a reduction in photosynthetic area and can potentially lead to infection of the fruit, causing crop failures and economic losses [18]. BLS may not reduce the fruit quantity, but the economic loss is mainly due to lower market value of the unwanted damaged fruit [19].

*Xanthomonas* biotypes causing BLS are widely distributed (Figure 3). Of these, *X. euvesicatoria* pv. *euvesicatoria*, *X. euvesicatoria* pv. *perforans*, *X. hortorum* pv. *gardneri* and *X. vesicatoria* biotypes causing BLS have been found in five continents (Africa, Asia, Europe, North America, and South America), while in Australia, *X. euvesicatoria* pv. *euvesicatoria*, *X. euvesicatoria* pv. *perforans* and *X. vesicatoria* have been isolated [20]. In the USA, it was found that predominant strains can change over time [21]. This has also been seen with other bacterial pathosystems, where pathogen populations appear to change in response to the release of host cultivars that were bred to be resistant to an earlier predominant strain [22].

Bacterial taxonomy has always been complicated, and *Xanthomonas* stands out as a seminal example of the challenges of bacterial nomenclature. A bacterium causing BLS was formally identified in South Africa as *Bacterium vesicatoria* [23]. In the same year, in Indiana, USA, a pathogen causing BLS was isolated from a tomato plant and named as *B. exitiosum* [24]. This was later reclassified as *X. vesicatoria* [25]. The pathovar convention was instituted in response to the poor biochemical resolution of multiple species of *Xanthomonas*, with the fact that their key distinguishing features were their host range. Thus, biochemically indistinguishable strains were placed into individual species, which were further demarcated by a pathovar epithet depending on the host [26]. Following this consolidation, the main known BLS pathogen was reclassified as *X. campestris* pv. *vesicatoria*.

Later DNA homology studies found that *X. campestris* pv. *vesicatoria* could be resolved into two distinct species: *X. axonopodis* pv. *vesicatoria* as type A and *X. vesicatoria* as type B [27]. Earlier still, another BLS pathogen on tomato, originally described as *Pseudomonas gardneri* [28] was reclassified as *X. gardneri* [29]. A fourth pathogen, provisionally placed in Group C, was formally described as *X. perforans* [29]. Thereafter, reclassification of the *Xanthomonas* species which causes BLS resulted in four species: *X. euvesicatoria* as group A; *X. vesicatoria* as group B; *X. perforans* as group C, and *X. gardneri* for group D [29]. The strain of *X. gardneri* was proposed for reclassification and named as *X. cynarae* pv. *gardneri* after similarity results of whole genome sequences between *X. gardneri* and *X. cynarae* were found to be above the 95–96% threshold [30]. The whole-genome based phylogeny, overall genome-related indices calculations and biochemical phenotypic profiling were obtained to reclassify the former pathovar of *X. cynarae* as *X. hortorum,* which resulted in a new name as *X. hortorum* pv. *gardneri* [31]. Thus, currently, four distinct *Xanthomonas* biotypes cause similar BLS symptoms on a range of solanaceous plants, and it is important to identify the specific strain to understand the epidemiology of the problem.

There are further pathotype divisions among the BLS strains. This is based on hypersensitive or susceptible reactions associated with four host resistance genes Bs1-4 [32] and corresponding bacterial effectors [33], which are presented in Table 1. While eleven pathotypes have been identified for strains within *X. euvesicatoria* pv. *euvesicatoria*, this assessment does not appear to have been completed for *X. euvesicatoria* pv. *perforans*, *X. hortorum* pv. *garnderi,* and *X. vesicatoria*. It is important to know what races are present in order to effectively manage them using genetic means.

## 3. Infection Process and Life Cycle

There are two critical infection processes for *Xanthomonas*. The first is the epiphytic phase where bacterial cells are introduced to aerial surfaces such as leaves and fruits [41]. After initial contact, the bacteria penetrate through natural openings such as stomata, wounds or hydathodes [42]. They then continue by moving to the center veins to multiply [43]. The second phase is the endophytic phase where the bacteria multiply within the host tissue. Subsequently, a high bacterial cell population develops, and the bacteria re-emerge onto the surface of the leaf and through rain and wind dispersal are transmitted into a new host and a new infection cycle is initiated [44]. When the bacteria reach a sufficient population, they invade the mesophyll tissues causing the disease symptoms on the leaves [43].

*Xanthomonas* employ important strategies to attach to the host plant and infect a leaf or fruit surface. These include the use of specific adhesins such as the XadM protein and lipopolysaccharides (LPS) [45]. These adhesins facilitate the attachment to the surface of the plant and have an important role in virulence expression of *Xanthomonas* [44]. The LPS provides protection from antimicrobial compounds and unfavorable environmental conditions, such as drying [46]. After attaching to the leaf, *Xanthomonas* forms a biofilm that is important for its survival [47]. Biofilms are three-dimensional structures composed of a bacterial community and its extracellular matrix, which attaches to substrates. The extracellular matrix consists of exopolysaccharides (EPS), extracellular DNA, protein, and lipids [48]. In *Xathomonas* species, the biofilm matrix also contains xanthan gum, production of which is directed by an operon of genes, including *gumB* to *gumM* [49,50]. The extracellular matrix is critical to the infection process.

Formation of the *Xanthomonas* biofilm is governed by intricate genetic processes. Responses of the bacteria to their population density, known as quorum sensing, are known to give better protection against biotic and abiotic factors [51]. Although quorum sensing aspects for BLS-inducing *Xanthomonas* species are not well characterized, those of the closely related *X. axonopodis* pv. *citri*, causal agent of citrus canker [52], provide insights. Quorum sensing in this pathogen induces flagellum production, triggering both swimming and swarming motility, followed by attachment of biofilm from planktonic cells resulting in a mature biofilm [52]. This strengthens the colonization of host tissue in canker induction. Given the broad utility of quorum sensing in closely related bacterial pathosystems, it is likely these also operate in BLS interactions.

*Xanthomonas* strains are reportedly able to survive from season to season in chilli and tomato seed, dead tomato debris, soil, and in the rhizospheres of soybean, tomato, and wheat [53]. BLS epidemiology is influenced by environmental factors which promote the broader dispersal of pathogens. This leads to the question of why susceptible plants are infected in some areas, while plants grown from the same seed in another area may not develop symptoms [41]. It was reported that *X. euvesicatoria* pv. *euvesicatoria* can survive in plant debris, inoculated sandy loam soil, and the rhizosphere of chilli and non-host plants. In addition, the pathogen was found to survive at least 18 months either in soil or in seed in a field previously planted with chilli diseased with BLS [53]. The pathogen can also be spread from infected plants to healthy plants by rain and wind. The life cycle of *Xanthomonas* causing BLS of chilli is presented in Figure 4.

## 4. Disease Detection

Several conventional and advanced methods have been used to diagnose BLS. Conventional detection is mainly based on visual identification of BLS symptoms, presence of bacterial oozing, and culture-based confirmation (Figure 1). *Xanthomonas* species can grow on a range of media, including Wilbrink’s media, Yeast-Glucose-Calcium Carbonate Agar (YGCA), Nutrient Broth-Yeast Extract Agar (NBYA), Adenine-supplemented Yeast Peptone Glucose Agar (YPGA), King’s B media, Nutrient Agar (NA) and Nutrient Dextrose (ND) [55]. While most *Xanthomonas* species are typically readily cultured, differentiating them can present challenges.

*Xanthomonas* species can be difficult to identify based on direct observations of cultures. With few exceptions, such as the pigmentless *X. axonopodis* pv. *mangiferaeindicae* of mango, and *X. fragariae*, which is more cream yellow than bright yellow [56], a given *Xanthomonas* species looks like any other *Xanthomonas* species. While host could be useful for pathovar determination in other plants, this cannot be done for BLS, as four distinct biotypes may occur on a single host. Protein profiles visualized using Sodium Dodecylsulfate Polyacrylamide Gel Electrophoresis (SDS-PAGE) were historically used to differentiate strains. *X. euvesicatoria* pv. *euvesicatoria* has a specific protein around 32–35 kDa, while *X. euvesicatoria* pv. *perforans*, *X. hortorum* pv. *gardneri*, and *X. vesicatoria* have one at 25–27 kDa [57]. Serological assays using a monoclonal antibody were also commonly used to detect *X. euvesicatoria* pv. *euvesicatoria* [58]. Fatty Acid Methyl Ester (FAME) analysis can also distinguish the *Xanthomonas* but requires specialized equipment and expertise [59]. Pectolytic and amylolytic activities are relatively simple methods that can distinguish BLS strains. *X. euvesicatoria* pv. *euvesicatoria* and *X. hortorum* pv. *gardneri* both have negative results for amylolytic and pectolytic activities, while *X. euvesicatoria* pv. *perforans* and *X. vesicatoria* strains strongly digest starch and pectic substrates [29]. The features of *Xanthomonas* species causing BLS of chilli are provided in Table 2.

Molecular methods have increasingly been adopted for BLS diagnosis. A range of PCR primers targeting different genes have been designed to detect different BLS *Xanthomonas* species. Since the 1980s, scientists have been using PCR for the detection of plant diseases. PCR has considerable advantages over other detection methods because it does not require isolation and cultivation of pathogens and is highly sensitive and faster than cultured-based methods. As a result, many PCR-based diagnostic methods for the identification of BLS pathogens have been reported. In 1994, the *hrp* gene clusters which determine the hypersensitivity and pathogenicity responses were selected to detect and identify *X. euvesicatoria* pv. *euvesicatoria* [62]. Oligonucleotide primers specific for *hrp* genes were used to detect the bacterium by PCR, and it was determined that as few as 10 CFU/mL could be detected. Furthermore, the pairs of Xeu2.4 and Xeu2.5 primers have been successfully developed to detect *X. euvesicatoria* pv. *euvesicatoria* and validated using 64 strains of the bacterium from different hosts and regions of origin. The assay sensitivity was up to 1 ng/mL of DNA, which made this protocol suitable for detecting the target in symptomless plants [63]. Primers XCVF and XCVR designed from the *rhs* family of proteins from *X. euvesicatoria* pv. *euvesicatoria* were developed with consistent results of amplification of a 517 bp fragment from various PCR machines and *Taq* polymerase enzymes provided by different manufacturers [64]. Amplified Fragment Length Polymorphism (AFLP) is another PCR-based technique which involves restriction of genomic DNA to detect polymorphisms in DNA. In 2009, four specific different primers pairs, Bs-XeF/Bs-XeR; Bs-XvF/Bs-XvR; Bs-XgF/Bs-XgR; and Bs-XpF/Bs-XpR, were developed to detect and differentiate all four *Xanthomonas* biotypes causing BLS using an AFLP based approach [65]. All strains from different geographic origins were identified and discriminated by the four different primer sets.

Even though PCR-based methods have become the gold standard for disease detection, including for *Xanthomonas* species, these all require a laboratory facility. Rapid field diagnosis can be achieved using Recombinase Polymerase Amplification (RPA) or Loop-Mediated Isothermal Amplification (LAMP). RPA has been developed and successfully used for field detection of BLS pathogens, including *X. euvesicatoria* pv. *euvesicatoria*, *X. euvesicatoria* pv. *perforans*, *X. hortorum* pv. *gardneri*, and *X. vesicatoria* [66]. Likewise, LAMP protocols have been adopted because they are quick, simple to perform and highly sensitive [67,68]. The sensitivity of LAMP was reported to be up to 1 pg/µL of genomic DNA, however, by calculation, this represents approximately 0.17 genomes per µL, so may be overstated. However, with a 30-min processing time and utility for field diagnostics, LAMP could become an important management tool for BLS [68]. A summary of primers which have been used to detect *Xanthomonas* causing Bacterial Leaf Spot of chilli is presented in Table 3.

## 5. Management and Control

Management for BLS is complicated because of an absence of resilient host resistance. Control mainly revolves around cultural practices, chemical control, host genetic resistance and, to a lesser extent, biological control. *Xanthomonas* species that cause BLS are known to remain in the seed, therefore the use of disease-free-seeds and transplants is one of the key management tools. Diagnostic testing of seeds and seed treatments are typically used to detect the level of seed infection [75]. After seed testing, a combination of planting resistant varieties, strict phytosanitation and bactericidal applications are control options for BLS disease [76].

### 5.1. Chemical Control

Copper products have historically been used as the main chemical prevention measure for BLS. The use of copper was started by Professor Millardet in 1882, who developed Bordeaux mixture (copper sulphate and lime), which was effective at controlling grape downy mildew [77]. The mixtures of copper and mancozeb were significantly effective for control of BLS compared to only basic copper sulphate [78]. In addition, copper-tolerant strains were killed by the combination of copper and mancozeb, while the sprays of fixed copper controlled only sensitive strains [79]. The addition of mancozeb was known to increase solubility of copper and led to improvements in the effectiveness of copper against tolerant *Xanthomonas.*

The use of small molecule compounds is an alternative approach to control BLS. Application of D-leucine and 3-indolylacetonitrile (IAN) effectively inhibits biofilm formation [80]. In addition, the application of IAN reduced BLS on tomato significantly and improved the effectiveness of copper against *X. euvesicatoria* pv. *perforans* [81]. Particularly, the regulation of BLS caused by copper-resistant *X. euvesicatoria* pv. *perforans* has been substantially improved by IAN. Carvacrol, the major component of essential oils from Zataria multiflora, also reported to be effective against plant pathogens, significantly decreased the severity of BLS on tomato and improves the efficacy of copper against copper-resistant *X. euvesicatoria* pv. *perforans,* which provides a more environmentally sustainable approach to control BLS [82]. The application of acibenzolar-S-methyl (ASM) treatment through weekly foliar sprays showed significant reductions of BLS in field-grown pepper [83]. ASM is a plant activator which induces systemic resistance in some plants, including pepper and tomato. The new strategy was developed using a hybrid nanoparticle, copper-zinc, on sensitive strains [84]. Based on this approach, compared to the controls, the hybrid nanoparticle significantly decreased the growth of bacteria in vitro. Several other small molecules also have been reported to inhibit the growth of BLS pathogens [85]. These inhibited the four BLS *Xanthomonas* biotypes but were also effective against copper- and streptomycin-resistant *Xanthomonas* strains. Importantly, there appear to have minimal impacts on the beneficial bacteria. These small molecules were composed by several amine-based functional groups, such as indole, imidazole, oxazole, piperidine, pyrazol, pyrimidine, and quinoline, which have been recognized for their antimicrobial activities [85].

### 5.2. Biological Control

Biological control is a promising alternative in disease management by inhibiting the growth of pathogens, improving plant defenses, or adjusting the environment to foster beneficial microbes. Biological control offers an environmentally sustainable alternative to traditional control methods that does not impact beneficial organisms [86] and is a low-cost strategy that can enjoy sustained utility if applied judiciously.

Biological control agents have been reported in several areas of chilli and tomato farming against Bacterial Leaf Spot, for instance, in vitro methods revealed that *Lactobacillus* MK3, *Trichoderma reseii* QM 9414, and *Pseudomonas aeruginosa* 1128 have showed significant potential as antagonists of *X. euvesicatoria* pv. *euvesicatoria* [87]. *Bacillus velezensis* IP22 was able to reduce the infected pepper leaves by 65% compared to positive control inoculated by *X. euvesicatoria* pv. *euvesicatoria*. This impact was related to the production of antimicrobial metabolites, lipopeptides from fengycin and locillomycin families, representing a promising strategy as a biocontrol agent of BLS of chilli [87]. *Bacillus pumilus* INR7 was confirmed to induce the resistance against *X. euvesicatoria* pv. *euvesicatoria* when combined with benzothiadiazole (BTH), a chemical inducer which boosted the expression of defense marker genes of pepper, CaPR1, CaTin1, and CaPR4 [88]. Significant reduction of approximately 50% in the severity of BLS was also obtained using humic acids and suspension (10^8^ CFU/mL) of *Herbaspirillum seropedicae* in tomato [89]. *Paenibacillus elgii* was evaluated and effectively suppressed the chilli BLS in pot trials with control values of 67% [90]. The mutant strain of *X. euvesicatoria* pv. *euvesicatoria* 75-3S hrpG was also evaluated for BLS under both greenhouse and field conditions and showed significant reductions in disease of 57% compared to control [91].

Bacteriophages are viruses which infect and replicate within bacteria [92]. These been studied as natural antimicrobial agents for BLS of chilli and tomato. A foliar application of bacteriophage decreased the BLS incidence from 40.5% (control) to 0.9% on greenhouse-grown tomato [93]. Ten bacteriophages were successfully isolated and characterized, aiming to contribute to an integrated BLS control strategy of reducing the use of conventional pesticides [94]. The other bacteriophage studies were also reported such as using Bacteriophage KΦ1 [95], XaF13 [96], ΦXaF18 [97] isolated from a pepper field infected by BLS. On a commercial scale, bacteriophage has been used to control BLS of chilli. One of the profitable bacteriophages has been produced by The AgriPhage [98] and showed that this bacteriophage management becomes another choice to deal with the BLS issue. The combination of biological control and other methods, such as systemic acquired resistance (SAR) inducer, was also reported to have promising results on BLS. The integration of bacteriophage, SAR, and copper hydroxide was able to reduce the disease intensity by 96–98% in pepper [99].

Some microbes proposed as prospective biocontrol agents behave as opportunistic pathogens which are included in the biosafety level 2 (BLS-2) classification. These pose risk not only to the environment, but to humans. For instance, *P. aeruginosa* is known to be promising for the control of BLS, but has adverse effects on humans, particularly immunocompromised patients such as those suffering from pneumonia, urinary tract infection and gastrointestinal infections [100,101]. Therefore, appropriate strategies to mitigate risks to humans and the environment need to be undertaken. It is advisable to carry out taxonomic characterization up to species and strain level, including whole genome sequencing. This aims to determine the possibility of the closeness of the target microorganism to BSL-2 or above [101]. Furthermore, the Environmental and Human Safety Index (EHSI), a tool to evaluate the safety of biocontrol candidates before being used in the field, can be adopted for biosafety awareness [102].

## 6. Host–Pathogen Interactions

There are currently more than 1200 genomes of *Xanthomonas* available (NCBI website). This will help in understanding taxonomy, pathogenicity, and virulence factors of *Xanthomonas* [41]. *Xanthomonas* sp. genomes encode more than 4,000 proteins [44]. The *Xanthomonas* strains causing BLS rely on some protein effectors to compete with other microorganisms and to colonize the chilli plant. These protein effectors are delivered to their targets by bacterial secretion systems. In *X. euvesicatoria* pv. *euvesicatoria,* it is known to have five different secretion systems, including Type I, II, III, V and VI [103]. Type 1 secretion system (T1SS) has a main role in quorum sensing signal [103]; T2SS in cell wall degradation [104]; T3SS maintains the successful of pathogenicity [105]; T5SS in plant attachment [106]; while T6SS is to compete with other microorganisms [107]. In all *Xanthomonas* species, there are two pathogenesis-correlated gene clusters; the first one is xps, which encodes type II secretion system (T2SS), and rpf, which coordinates the synthesis of pathogenicity factors.

The type II secretion (T2S) system primarily secretes degradative enzymes transported by the Sec or TAT (twin-arginine translocation) systems through the inner bacterial membrane and also contributes to the virulence of plant pathogenic bacteria. Various *Xanthomonas* spp. contain two T2S systems encoded by homologous gene clusters of xps and xcs. A virulence function has been identified for xps-T2S systems and appears to be dispensable for virulence [108]. The Xps-T2S system from the plant pathogen *X. vesicatoria* promotes disease and leads to the translocation of the type III secretion (T3S) mechanism of proteins effectors that are released into the plant cell [109]. Thus, understanding the T3 effectors is critical to understanding host and pathogen interaction [110].

The Xps-T2S system secreted degradative enzymes such as cellobiosidases, cellulases, lipases, endoglucanases, polygalacturonases, xylanases, and proteases [111]. The rpf gene is another gene cluster that plays an important role in the pathogenicity of *Xanthomonas*. Rpf is a regulation of pathogenicity factors present in all Xanthomonads that encodes components that control the synthesis and perception of the signal molecule DSF (diffusible signal factor). The synthesis of the virulence factors and biofilm formation is controlled by the rpf of the DSF systems, and it is necessary for full virulence of *X. euvesicatoria*, *X. oryzae* pv. *oryzae*, *X. oryzae* pv. *oryzicola*, *X. campestris* pv. *armoracle* [106].

The synthesis of extracellular enzymes, extracellular polysaccharide biofilm dispersal and virulence in *X. euvesicatoria* pv. *euvesicatoria* is positively controlled by the rpf gene cluster [112]. Some of the rpf genes have been described in detail. The major *X. euvesicatoria* pv. *euvesicatoria* aconitase is encoded by rpfA that involved in iron homeostasis. In the synthesis of a small diffusible signal molecule, rpfB and rpfF are involved [113].

## 7. Type III Effector Biology and Action

Suppression of plant immunity is an effective virulence strategy to colonize the host by Gram-negative bacteria. The Gram-negative bacteria use a type III secretion mechanism to provide effectors into the host cell to accomplish this. These effectors target the main plant immune signaling pathway modules [114]. This type III secretion system (T3SS) is controlled by the hypersensitive response and pathogenicity (hrp) gene cluster [111]. There are more than 20 genes in this gene cluster, grouped into various transcriptional units [111]. Avirulence proteins are a class of type III effectors in plant pathogens. The term “avirulence” (avr) describes bacterial genes that specify the basic recognition of bacteria by plants with a resistance (R) gene that matches them. The detection of an avr protein mediated by the plant R gene leads to the induction of a plant defense reaction that typically involves hypersensitive response (HR), a rapid localized cell death associated with arresting pathogen ingress. Avirulence protein limits the host range of the pathogen, an effect that is detrimental to the bacteria and is likely not their primary role [115].

The AvrBs3 protein is a broad family of proteins retained in the *Xanthomonas* genus, which also has a role in both avirulence and virulence. The effector of AvrBs3 from *X. vesicatoria* on chilli and tomato is a causal agent of BLS. Symptoms of early infection are likely to coincide with the release of water and nutrients that support bacterial multiplication. The ability of *X. euvesicatoria* pv. *euvesicatoria* to cause disease in susceptible plants and to induce HR in resistant plants depends on the T3SS encoded by hrp gene cluster [116]. The hrp gene cluster encodes more than 20 proteins, 11 of which are retained in the pathogenic bacteria of plants and animals. These genes have been renamed hrc (hrp conserved) and are assumed to encode the core components of the secretion apparatus. Expression of hrp gene in *Xanthomonas* is regulated by HrpG and HrpX. As an OmpR family regulator, HrpG triggers the hrcC and hrpX expression in *X. vesicatoria* or only hrpX in *X. euvesicatoria* pv. *euvesicatoria*. The expression of other hrp genes along with some effector genes is regulated by hrpX, an AraC-type transcriptional activator [117]. The mechanistic insight of T3SS in *X. euvesicatoria* pv. *euvesicatoria* is explained in Figure 5.

## 8. Resistance Genes

Host-plant resistance is a significant component of an integrated management program for BLS. There are five known independent dominant genes (Bs1, Bs2, Bs3, Bs4, and Bs7) and two recessive genes (Bs5 and Bs6) for qualitative resistance controlling BLS resistance [30,106,128]. A combination of Bs1, Bs2, and Bs3 genes gave the lowest disease scores of BLS. A transgenic approach to disease resistance was investigated in tomatoes based on the chilli resistance gene Bs2 [129]. For the Bs2 gene, the avirulence gene avrBs2 was reported to be involved in the fitness of *X. euvesicatoria* pv. *euvesicatoria* and is known to be highly conserved among other *Xanthomonas* species. This finding also suggested that Bs2 gene in other plant species is functional, which might support stable resistance [130].

The Bs3 gene in chilli also conferred resistance to *X. vesicatoria*. It has been shown in genetic and molecular studies that *X. euvesicatoria* pv. *euvesicatoria* strains expressing the AvrBs3 gene trigger Bs3-mediated resistance [116]. AvrBs3 encodes the AvrBs3 family type member, a large family of bacterial effectors that have a sequence identity of 80–99%. In their C-terminus, the AvrBs3-like proteins contain nuclear localization signals (NLSs) and a transcriptional activation domain (AD) that are critical to their virulence [131]. Two recessive genes in chilli, Bs5 andBs6, conferred resistance to all races of Bacterial Leaf Spot of chilli. These recessive genes were reported as more resilient than the dominant resistance genes, as they do not require complex gene-for-gene interactions [132]. The impact of two recessive genes, Bs5 andBs6 genes, showed thatBs5 confers a higher resistance level thanBs6 at 25 °C, but they confer higher resistance to P6 (*X. euvesicatoria* pv. *euvesicatoria* chilli strain XV157 of race 6) in combination, suggesting at least the action of the additive gene [133].

## 9. Genes Involved in Plant Defenses

Several defensives signaling mechanisms have been formed by plants to protect them from the adverse conditions of environment and pathogen attack. The effectiveness of phenylalanine ammonia-lyase (PAL) extracted from pepper (CaPAL1) in defense responses to pathogens is involved in salicylic acid (SA) biosynthesis, an important signal involved in plant systemic resistance [134]. The chilli leaf was induced with CaPAL1 gene by avirulent *X. euvesicatoria* pv. *euvesicatoria* infection. There was an increased vulnerability to virulent and avirulent *X. euvesicatoria* pv. * euvesicatoria* infection in CaPAL1-silenced chilli plants.

The CaCYP1 gene expression (cytochrome P450 from *Capsicum annuum* L. *Bukang*) was found after leaf hypersensitive reaction by infection of chilli plants with the non-host pathogens *X. axonopodis* pv. *glycines* [135]. The results suggested that in plant defense response pathways involving salicylic acid and abscisic acid signaling pathways, CaCYP1, a new cytochrome P450 in chilli plants, may play a role. In the plant defense response against pathogens, SA and ethylene are essential secondary signals. A cytoplasmic protein kinase (RLCK), receptor-like chilli (*Capsicum annuum*) gene (CaPIK1) was identified and transcriptionally activated by infection with *X. euvesicatoria* pv. *euvesicatoria* [136]. In chilli plants, silencing of CaPIK1 confers increased susceptibility to infections with *X. euvesicatoria* pv. *euvesicatoria*. The results indicate that CaPIK1 modulates the signalling needed for the defense response to pathogen infection based on salicylic acid. In chilli, the CaPIK1 gene was transcriptionally induced by virulent and avirulent *X. euvesicatoria* pv. *euvesicatoria* infection, leading at an early stage of infection to a high level of gene expression. These results support the possibility that CaPIK1 gene can confer as a signal transduction mediator in chilli plants’ defense responses to pathogen invasion.

## 10. Conclusions and Prospects

The challenges in BLS of chilli are to further develop detection platforms, determine the factors which account for the pathogenesis in specific hosts or strains, and include resistance in the breeding program. Further mapping of the prevalent strains is essential to improve management responses. The application of functional genomics and proteomics will help to identify the genes and proteins involved in the infection lifecycle of the different *Xanthomonas* species and strains. Mapping the genes that govern host–pathogen interactions will provide targets for future application of gene-editing technologies. These actions need to be integrated across the value chain from researchers through to farming stakeholders to meet the demand for this iconic fruit.

## Figures and Tables

**Figure 1 plants-11-02291-f001:**
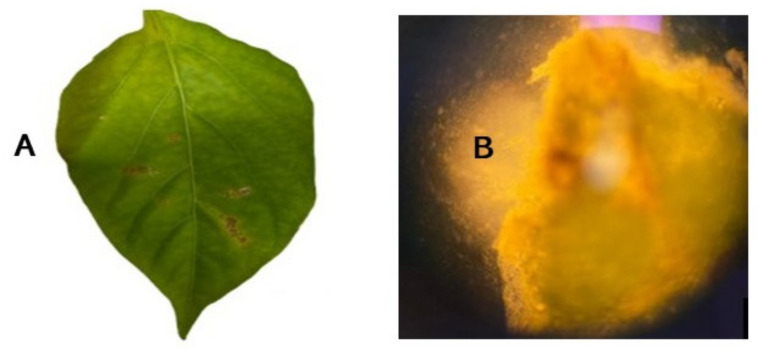
Diseased chilli leaf infected with *X. euvesicatoria* pv. *euvesicatoria* (**A**), and bacterial oozing from lesion viewed under phase contrast (100× magnification) (**B**).

**Figure 2 plants-11-02291-f002:**
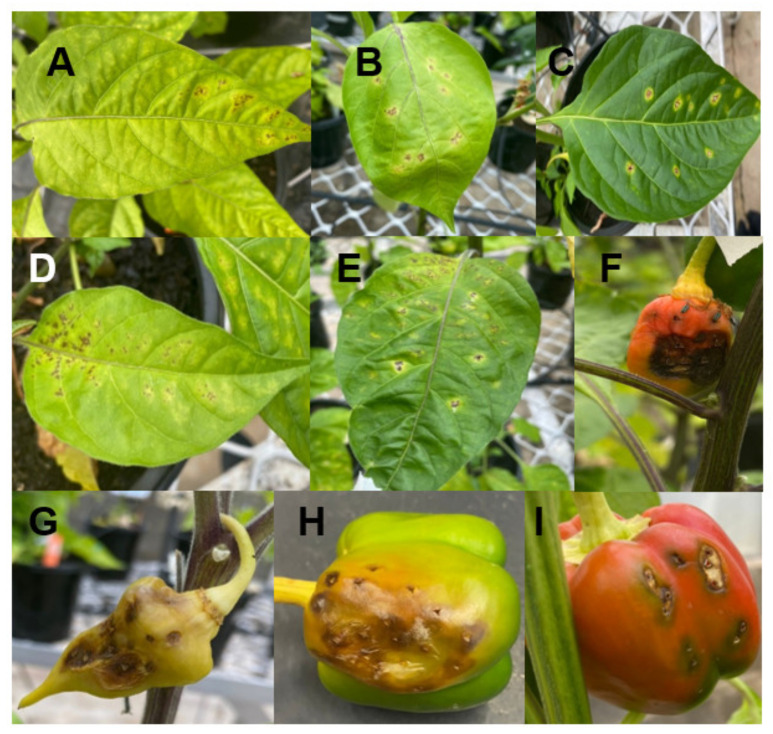
BLS symptoms on leaves (**A**–**E**) and fruits of different chilli cultivars, *Capsicum chinense* Jacq (**F**,**G**) and *Capsicum annuum* (**H**,**I**).

**Figure 3 plants-11-02291-f003:**
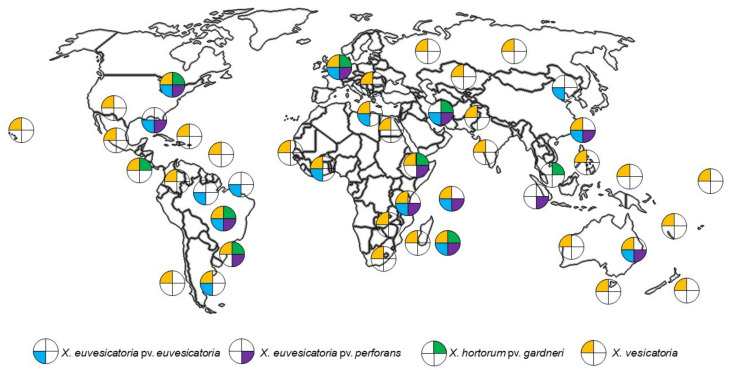
Current distribution of *Xanthomonas* spp. causing Bacterial Leaf Spot of Chilli. The database represents records between 2010 to 2020 of *Xanthomonas* was obtained from European and Mediterranean Plant Protection Organization (EPPO) via https://gd.eppo.int/taxon/XANTEU; https://gd.eppo.int/taxon/XANTGA; https://gd.eppo.int/taxon/XANTPF; https://gd.eppo.int/taxon/XANTVE accessed on 1 April 2022. The world map was purchased from https://www.etsy.com/au accessed on 1 April 2022.

**Figure 4 plants-11-02291-f004:**
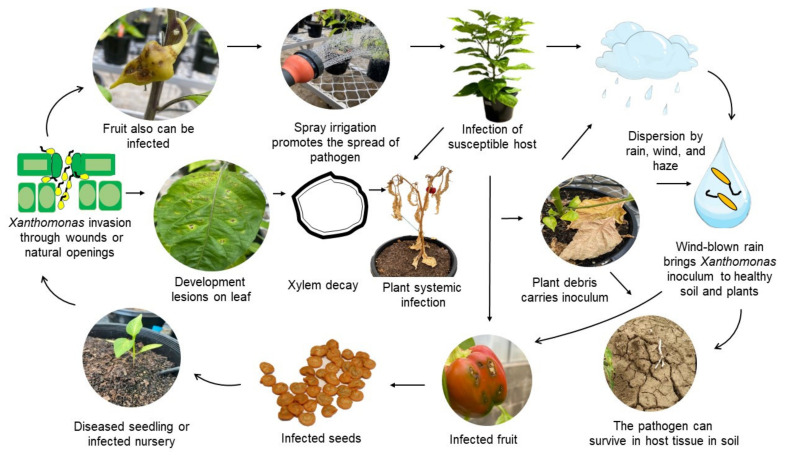
Disease cycle of Bacterial Leaf Spot of Chilli. Adapted from An et al. [41] and Osdaghi et al. [54] with some modification and addition.

**Figure 5 plants-11-02291-f005:**
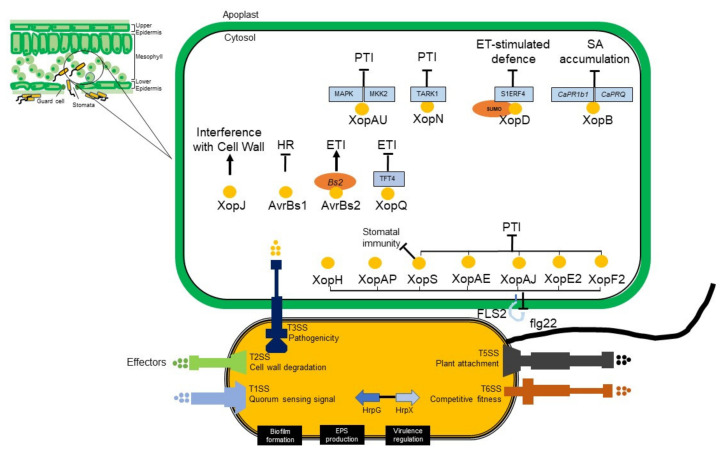
The mechanistic insight of the Type 3 secretion system in *X. euvesicatoria* pv. *euvesicatoria.* T3SS as to suppress the plant immunity, some effectors have been reported to promote the disease colonization. Some effectors have more than one role, such as XopF2, XopE2, XopAJ, XopAE [118], and XopS [119]. These effectors inhibit the flg-22 induced signalling and suppress the pattern-triggered immunity (PTI). Effectors XopAP and XopH are inhibitors of flg22-induced reporter gene activation, but the PTI-related inhibition is not known yet [107]. XopS has been reported also to stabilize pepper to regulate the defense response, stomatal immunity [119], and disease symptoms [120]. While effector AvrBsT suppresses the hypersensitive response (HR) as effect of effector protein AvrBs1 in resistant chilli plant [121]. Chilli plants carrying the Bs2 gene are resistant to *X. euvesicatoria* pv. *euvesicatoria* strains, which contain AvrBs2. The interaction between Bs2 and AvrBs2 resulted in Effector-Triggered Immunity (ETI), which led to suppression of effectors into plant cell [122]). During the infection, XopD effector is required for pathogen growth and symptom-development delay. This effector carries small ubiquitin-like modifier (SUMO) and desumoylates S1ERF4 to suppress ethylene levels, which increases susceptibility of host plant to *X. euvesicatoria* pv. *euvesicatoria* [123]. To interfere the host immune signalling, XopAU effector manipulates the host MAPK signal and activation of MKK2 [124]), while XopB interferes the PTI and suppresses SA accumulation [125]. XopN interacted with the Tomato Atypical Receptor-Like Kinase1 (TARK1) and suppresses PAMP-triggered immune response [126]. While XopJ is to inhibit the cell wall-based defense response [127].

**Table 1 plants-11-02291-t001:** Races and species of *Xanthomonas* causing Bacterial Leaf Spot.

Species	Races	References
*X. euvesicatoria* pv. *euvesicatoria*	0–10	[34,35,36,37,38,39,40]
*X. euvesicatoria* pv. *perforans*		
*X.**hortorum* pv. *gardneri*		
*X. vesicatoria*		

**Table 2 plants-11-02291-t002:** Differentiation of *Xanthomonas* spp. causing BLS of chilli.

	*X. euvesicatoria* pv. *euvesicatoria*	*X. euvesicatoria* pv. *perforans*	*X. hortorum* pv. *gardneri*	*X. vesicatoria*
Group	A	C	D	B
Reference strains	ATCC11633^T^ (NCPPB2968	ATCC BAA-983^T^ (NCPPB4321)	ATCC19865^T^ (NCPPB881)	ATCC35937^T^ (NCPPB422)
Protein unique size (kDA)	32–35	25–27	25–27	25–27
Amylolytic activity	-	+	-	+
Pectate hydrolysis	-	+	-	+
mAb * reaction	1, 21	30	8	8, 15
Utilization of:				
Dextrin	+	+	-	+
Glycogen	+	v	-	v
N-acetyl-D-glucosamine	+	+	-	v
D-galactose	+	+	-	v-
Gentibiose	+	+	-	v
α-D-lactose lactulose	v	+	-	v-
Acetic acid	v	+	-	-
Cis-aconitic acid	+	v	-	-
Malonic acid	+	+	-	v
Propionic acid	v-	+	-	v
D-alanine	v	+	-	v
Glycyl-L-aspartic acid	-	+	-	v-
L-threonine	v	+	-	v-

* Monoclonal antibodies developed using *X. euvesicatoria* pv. *euvesicatoria* strains which reacts in enzyme associated immunosorbent assay (Reproduced from Jones et al., 2004). + = positive reaction by all strains; v = 50% or more strains used the compound; v- = less than 50% of strains used the compound; - = none of strains. Refs. [29,54,57,60,61].

**Table 3 plants-11-02291-t003:** Summary of Molecular Detection on *Xanthomonas* causing Bacterial Leaf Spot Disease of Chilli. PCR = Polymerase Chain Reaction; LAMP = Loop-Mediated Isothermal Amplification; RPA = Recombinase Polymerase Amplification; AFLP = Amplified Fragment Length Polymorphism; MLSA = MultiLocus Sequence Analysis.

Bacterium	Target	(Assay)Primer Name	Sequence 5′-3′	Amplicon Size	Ref.
		**(PCR)**			
*X. euvesciatoria* pv. *euvesicatoria*	Zn-dependent oxidoreductase	ZnDoF	GGTGACAAACCGTCAGGAATAG	100 bp	[69]
		ZnDoR	CGCACTGGCACGTTATCA		
		**(LAMP)**			
*X. hortorum* pv. *gardneri*	*hrpB*	F3	CGGGGTGCAGGTCAGC	a/n	[68]
		B3	ACCGGCACCGCCAAG		
		FIP	CCACCTCGGCACGTTGCAGGCGAGGTATGCGAGTTGC		
		BIP	GCCGCCATCTCGCCTTGCGCCCCGATCCGATCACG		
		LB	CGAGCTGGTGGGCTTGT		
		**(RPA)**			
*X. hortorum* pv. *gardneri,**X. euvesicatoria* pv. *euvesicatoria,**X. euvesicatoria* pv. *perforans,**X. vesicatoria*	*hrcN*	*X. hortorum* pv. *gardneri* exo probe	TCTCGCCTTGCTGGCGCCGTTTGGCGAGC-dT-FAM-HG-dT-BHQ1-GGGCTTGTCGCGC	a/n	[66]
		XGF	GCACGCTGTTGCAACGTGCCGAGGTGGTGG		
		XGR	CGTCCGCCGGCTCACCCAGGCCATCGAGTA		
		*X. euvesicatoria* pv. *euvesicatoria* pathovar exo probe	CGGGCAAGGCGCAATCGCCTGTGACACC-dT-FAM-GHG-dT-BHQ1-GCCGATCCAGGCG		
		*X. euvesicatoria* pv. *perforans* exo probe	CGGGCAAGGAGCCATCGCCTGTGACACC-dT-FAM-GHG-dT-BHQ1-GCCGATCCAGGCG		
		FP1	GTTGGACCGGCCTTGCTGGGCCGCGTGCTC		
		RP1	GTCGGCATGGGGTGTTCGATCAGCCGCCGA		
		*X. vesicatoria* exo probe	GCCTTGCTGGCGCCGTTTGGCGAATTGG-dT-FAM-GHGG-dT-BHQ1-TGTCGCGCGAAA		
		XVF	ATGGCACGCTGTTGCAGCGCGCCGAGGTGGTGG		
		XVR	GCACCGCCAATGGGCGACCGGATCCGATCA		
		**(LAMP)**			
*X. euvesicatoria* pv. *euvesicatoria*	ATP-dependent DNA helicase (*recG*)	XeRec-F3	CCATGTAGGGCTTGTTGACG	a/n	[67]
		XeRec-B3	GGTGGTCGCATCTTCATTGG		
		XeRec-FIP	ACCCGCTCACGGAAAACGTGCC-TTCAGCGATGGACAGC		
		XeRec-BIP	GAGGCCACGTTGGCGATGAG-GTGAACGACGACGGTTCG		
		XeRec-LF	ACCCGGCAGGCACGGTGCT		
		XeRec-LB	AGCAACGTCGGCGCCGGATA		
		**(PCR)**			
*X. vesicatoria*	Ferric uptake regulator (*fur*)	fur1	GAATTCATCGGTCCTGGGAGTC	1572 bp	[70]
		fur2	AAGCTTCGGCGTGGAAGTGA		
		**(PCR)**			
*X. vesicatoria*	ATP synthase	XV1F	CAGTCCTCCAGCACCGAAC	365 bp	[71]
		XV1R	TCTCGTCGCGGAAGTACTCA		
		**(PCR)**			
*X. euvesicatoria* pv. *euvesicatoria*	XCV0215	XV4F	ATCAATGAGCCTTGGGATGTGACGA	230 bp	[72]
		XV4R	GCATAGGTCAGGGCTTGCTTTAGCG		
	XCV0217	XV5F	GCCTAAGAATGCGGAGCCTTGGCT	210 bp	
		XV5R	ATCTTCGGAGGCGTGTACGGCGTA		
	XCV3374	XV6F	AATGTGATCTTTTTGACGAGCGCA	169 bp	
		XV6R	GCAACCTCGTCTGTTTCATTCTCAT		
	XCV3818	XV7F	CATTTCCATCACGCGTCATGCCG	179 bp	
		XV7R	TGTTGCTCGGAATCGGTGGACCACC		
	XCV3902	XV8F	TGTCTCAAGCCGCGCTTAAC	123 bp	
		XV8R	AACCGAAGAACAGGAACGATCTC		
	XCV0217	XV10F	GCGTTGGCACAATGTCGACC	805 bp	
		XV10R	TTCGTCTAGCTCTCCACGGACCTG		
	XCV0655	XV11F	GCGACTGCGCTGGTATGAGCTCTA	631 bp	
		XV11R	TGGCGTGTAGACACCCACTGTCGAG		
	XCV1116	XV12F	GGAGCCGTCTGCTGGTAAGCTGAT	638 bp	
		XV12R	GCTGTATCAAACGAGATCCGCTG		
	XV1853	XV14F	TGGTTCACGTCATCGTTGTCGGA	713 bp	
		XV14R	TAGAGCTCGCTCAAAGCCCTTCGG		
		**(MLSA-AFLP)**			
*X. euvesicatoria* pv. *euvesicatoria*	*atp*D	atpD-F	GGGCAAGATCGTTCAGAT	756 bp	[73,74]
		atpD-R	GCTCTTGGTCGAGGTGAT		
		**(AFLP-PCR)**			
*X. euvesicatoria* pv. *euvesicatoria*		Bs-XeF	CATGAAGAACTCGGCGTATCG	173 bp	[65]
		Bs-XeR	GTCGGACATAGTGGACACATAC		
*X. vesicatoria*		Bs-XvF	CCATGTGCCGTTGAAATACTTG	138 bp	
		Bs-XvR	ACAAGAGATGTTGCTATGATTTGC		
*X. hortorum* pv. *gardneri*		Bs-XgF	TCAGTGCTTAGTTCCTCATTGTC	154 bp	
		Bs-XgR	TGACCGATAAAGACTGCGAAAG		
*X. euvesicatoria* pv. *perforans*		Bs-XpF	GTCGTGTTGATGGAGCGTTC	197 bp	
		Bs-XpR	GTGCGAGTCAATTATCAGAATGTGG		
		**(PCR)**			
*X. euvesicatoria* pv. *euvesicatoria*	*rhs family*	XCVF	AGAAGCAGTCCTTGAAGGCA	517 bp	[64]
		XCVR	AATGACCTCGCCAGTTGAGT		
		**(PCR)**			
*X. euvesicatoria* pv. *euvesicatoria*	hypothetical protein XCV3137	Xeu2.4	CTGGGAAACTCATTCGCAGT	208 bp	[63]
		Xeu2.5	TTGTGGCGCTCTTATTTCCT		
		**(PCR)**			
*X. euvesicatoria* pv. *euvesicatoria*	hypersensitive reaction and pathogenicity (*hrp*)	RST9	CACTATGCAATGACTG	355 bp	[62]
		RST10	AATACGCTGGAACTGCTG		

## Data Availability

Not applicable.

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
