# Peer review of "A Pan-Global Study of Bacterial Leaf Spot of Chilli Caused by Xanthomonas spp."

_plants, 2022, doi:10.3390/plants11172291_

Round 1

Reviewer 1 Report

The paper is well-organized. The topics are exhaustively explained. I loved the historic notes about the introduction of pepper as a crop.

I have few minor suggestions/edits:

1) Recently,  X. gardeneri, X. euvesicatoria and X. perforans have been reclassified as X. hortorum pv. gardeneri (1) X. euvesicatoria pv. euvesicatoria (2) and X. euvesicatoria pv. perforans (2),  respectively.

Authors might consider to use the most recent taxonomic classification

    (1)
  • DOI: 10.1099/ijsem.0.003104
  •  

(2) doi.org/10.1111/ppa.12461

2) For the detection methods, authors should include the qPCR assay published in 2016 by Strayer et al. (https://doi.org/10.1094/PDIS-09-15-1085-RE). 

The RPA assay is an "adaptation" of the qPCR assay previously developed by the same authors. Both methodologies are equally reliable and worth to be cited.

3) For the small molecules paragraph, authors should consider to include the work from Srivastava et al. (2021).  In their work they also included Xanthomonas strains isolated from pepper (https://doi.org/10.1094/PHYTO-08-20-0341-R)

No further comments or suggestions

Author Response

Re: Response to Reviewer 1 for plants-1808537

Thank you for considering our manuscript for publication. My co-authors and I believe we have adequately addressed your and the reviewers’ comments in our updated manuscript. For ease of processing, we can provide the following table that systematically addresses these comments and details changes made to the manuscript. Where we have accepted the suggested amendment, we have used ‘Changed’ or ‘Added’. We would appreciate your consideration of our responses in these cases and make a judgment call as to whether further amendments are warranted. Please do not hesitate to contact me should you require further information.

Comment 1.

Changed.

The most recent taxonomic classification has been adjusted in all the text.  The story of reclassification of X. hortorum pv. gardneri has been added in Line 119-131 with some additional papers for references.

Comment 2

Added.

We have included this paper in the table 3 “Summary of Molecular Detection on Xanthomonas causing Bacterial Leaf Spot Disease of Chilli. PCR= Polymerase Chain Reaction; LAMP= Loop-mediated isothermal amplification; RPA=Recombinase Polymerase Amplification; AFLP= Amplified Fragment Length Polymorphism; MLSA= MultiLocus Sequence Analysis”.

Comment 3

Added.

We have added this information in line 315-321.

Please do not hesitate to contact me should you require further information.
Thank you very much. 
Kind regards, 
Desi Utami

Reviewer 2 Report

1. The title should be more precise: "A review of the status..." which status??

Also, the authors should add "...: A pan-global study" in the title.

2. Abstract line 20: "Some biological controls..." please replace with "Some biological control agents"

3. Introduction L28: "Chilli (Capsicum spp.) is one of the most valuable spices in the world" no it is not, in terms of economic value Saffron (Crocus sativus) is the most valuable spice. 

4. L 28: please replace "alco" with "also"

5. L33: "The year 1492 saw the voyage of Christopher Columbus." The sentence is incomplete please club with the following sentence.

6. L53-56: The production data for 2020 is also available, kindly include the revised data.

7. Figure 1c and 1d: the purpose of including the "c) Negative result on amylolytic activity by X. euvesicatoria and (d) Positive result on amylolytic activity by X. vesicatoria" is not clear. I recommend excluding these figures

8. Figure 3: Direct link should be provided showing "Current distribution of Xanthomonas spp" not the link to EPPO webpage. Also the time period for this data should be provided.

9. Figure 4: A citation should be provided from where the original cycle was adopted

10. Sections 3 and 4 are very brief and do not preset the recent developments in the area. 

11. In subsection 5.2 "Biological control": It is highly recommended to include few sentences on the biosafety of biocontrol agents as the antagonists like Pseudomonas aeruginosa (L-268) and Bacillus velezensis (L-269) pose serious risk to human health. Please refer to https://doi.org/10.1016/j.scitotenv.2019.07.046

12. A model representing the mechanistic insights of "Host-Pathogen Interactions" with the information presented in sections 6, 7, 8 and 9 would add to the novelty of the review.

Author Response

Re: Response to Reviewer 2 for plants-1808537

Thank you for considering our manuscript for publication. My co-authors and I believe we have adequately addressed your and the reviewers’ comments in our updated manuscript. For ease of processing, we can provide the following table that systematically addresses these comments and details changes made to the manuscript. Where we have accepted the suggested amendment, we have used ‘Changed’ and ‘Added’. We would appreciate your consideration of our responses in these cases and make a judgment call as to whether further amendments are warranted. Please do not hesitate to contact me should you require further information.

Kind regards,

Desi Utami

  1. Changed

The title has been changed into:

“A pan-global study of Bacterial Leaf Spot of Chilli caused by Xanthomonas spp.”

  1. Changed

The sentence has been amended.

  1. Changed

We have changed the sentence.

  1. Changed.

We have replaced the word.

  1. Changed

The sentence has been amended.

  1. Changed

The data has been amended with 2020 data.

  1. Changed

The Figure 1c and 1 d have been deleted

  1. Added

The direct links from each Xanthomonas species have been added into the figure legend. The time period for the reporting data have been also provided.

  1. Added

The citation has been added to the text.

  1. Added

The additional information have been added from line 233 to line 257.

  1. Added

We are aware about the biosafety issue. We have added this paper along with some other relevant papers to explain about the biosafety of biocontrol agents in line 365-376.

  1. Added

We have added the model of Host-Pathogen Interactions in figure 5.

Round 2

Reviewer 2 Report

NIL

Author Response

There is no comment or suggestion from Reviewer 2 at this stage, therefore there is no response.